# Out-of-equilibrium dynamics of two interacting optically-trapped particles

Victor S. Dotsenko[1], Alberto Imparato[2], Pascal Viot[1] and Gleb Oshanin[1]

**1** Sorbonne Université, CNRS, Laboratoire de Physique Théorique de la Matière Condensée (UMR CNRS 7600), 4 Place Jussieu, 75252 Paris Cedex 05, France
**2** Department of Physics and Astronomy, University of Aarhus Ny Munkegade, Building 1520, DK–8000 Aarhus C, Denmark

## Abstract

We present a theoretical analysis of a non-equilibrium dynamics in a model system consisting of two particles which move randomly on a plane. The two particles interact via a harmonic potential, experience their own (independent from each other) noises characterized by two different temperatures $T_1$ and $T_2$, and each particle is being held by its own optical tweezer. Such a system with two particles coupled by hydrodynamic interactions was previously realised experimentally in Bérut et al. [EPL 107, 60004 (2014)], and the difference between two temperatures has been achieved by exerting an additional noise on either of the tweezers. Framing the dynamics in terms of two coupled over-damped Langevin equations, we show that the system reaches a non-equilibrium steady-state with non-zero (for $T_1 \neq T_2$) probability currents that possess non-zero curls. As a consequence, in this system the particles are continuously spinning around their centers of mass in a completely synchronized way - the curls of currents at the instantaneous positions of two particles have the same magnitude and sign. Moreover, we demonstrate that the components of currents of two particles are strongly correlated and undergo a rotational motion along closed elliptic orbits.

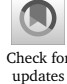
# 1  Introduction

Within the recent years there was much interest in stochastic dynamics of out-of-equilibrium multicomponent systems, different components of which are connected to thermostats kept at different temperatures. On the theoretical side, several minimalistic (albeit experimentally-realisable) models have been worked out, providing deep insights into the general aspects of an emerging non-trivial and sometimes even a counterintuitive dynamical behavior. Such models were also used as a framework for checking the validity of various fluctuation relations and theorems [1–8] and also for justifying the notion of effective temperatures [8, 9]. A few stray examples are the Brownian gyrator model [10, 11] and its various generalizations [12–23], models of interacting particles connected to different heat baths [24, 25], models of the directional influence between cellular processes [26], coupled Kuramoto oscillators kept at different temperatures [27], bead-spring models [28–30] and the molecular "spinning tops" in two-dimensional systems [31]. A common feature of several of these theoretical models is that they exhibit a motor effect, in the form of particle translational or rotational motion, as a consequence of both the broken spatial symmetry and the lack of thermal equilibrium.

On the experimental side, the behavior predicted by the theoretical analysis of the Brownian gyrator model has been validated experimentally. It was done by either constructing equivalent electric circuits [13, 14], or by studying directly the dynamics of a Brownian colloidal particle that is optically trapped in an elliptical potential well and is simultaneously coupled to two heat baths kept at different temperatures acting along perpendicular directions [32]. Similarly, such out-of-equilibrium systems were experimentally realised in a single-electron box consisting of two islands with a tunnel junction [33] and with two optically-trapped viscously coupled particles, in contact with two effective baths maintained at different temperatures [34, 35].

The experimental set-up in [34, 35] consists of a disc-shaped cell (with 18 mm in diameter and 1 mm in depth) in which there are two suspended micrometer-sized beads - 1 and 2 - that are confined by optical tweezers centered at two distinct spatial positions (see also [36] for a similar set-up) at distance $15\,\mu m$ above the lower surface of the cell and some distance $2x_0$ apart of each other. The two beads are experiencing two different effective temperatures - $T_1$ and $T_2$, respectively: this crucial condition is experimentally realised in [34, 35] by adding a Gaussian white noise to the position of either of the tweezers. As shown in [34], once the amplitude of the displacement is sufficiently small to ensure the validity of a linear regime, such an additional random force does not affect the stiffness of the tweezer (which therefore remains constant) but merely increases the effective temperature. Lastly, in such a set-up the beads are hydrodynamically coupled to each other; that being, they interact between themselves through the motion of a surrounding viscous fluid. Formulating the model in terms of coupled Langevin equations for the positions of the beads and introducing the forces through the Rotne-Prager diffusion tensor, it was demonstrated in [34, 35] (see also the earlier [36] for the analysis in the $T_1 = T_2$ case) that the inter-bead interaction is elastic, i.e., is a quadratic function of the instantaneous distance between the beads, and the proportionally factor in this function is dependent in the leading order only on the fixed distance $2x_0$ between the centers of the optical traps. A comparison of the solutions against an experimental data has shown that such an approximation is quite accurate. Clearly enough, this picture is only valid for sufficiently stiff traps such that the beads do not travel far away from the centers of their respective optical traps. For "loose" traps this is not the case, and this is not the case either in situations when the distance $2x_0$ becomes large and the beads get effectively decoupled from each other.

The theoretical analysis in [34, 35] focused on the behavior of the effective heat fluxes between the two beads in the out-of-equilibrium state with $T_1 \neq T_2$. It was demonstrated that these fluxes obey an exchange fluctuation theorem in the stationary state and moreover,

the total hot-cold flux satisfies a transient exchange fluctuation theorem at any time, while the total cold-hot flux obeys this theorem only at large enough times. However, these conceptually important results were derived under an assumption that the stochastic dynamics of the two particles can be viewed as an effectively *one-dimensional* process that evolves along the line connecting the centers of two optical traps. Within such an assumption, the model becomes mathematically equivalent to the bead-spring model considered in [28, 29] or the Brownian gyrator model with an external forcing [19, 37]. Then, a legitimate question is whether due to such a restriction some remarkable features of the dynamical behavior are overlooked.

In the present paper, motivated in part by the "spinning tops" model put forth in our recent paper [31], we revisit the dynamical behavior in the system considered in [34, 35], allowing now the beads to move on a plane, which is somewhat closer to the actual geometrical set-up. Apart from the additional spatial dimension, our model here remains essentially the same as the one formulated in [34, 35]: Each bead it optically trapped by its tweezer and the temperatures $T_1$ and $T_2$ at which the particles live are not equal to each other. We proceed to show that the dynamical behavior is indeed much more complex than in the 1D case: In fact, it appears that the two beads undergo a completely synchronized spinning around their centers of mass due to a systematic torque exerted on the particles. The term "completely synchronized" here means that not only the sign but also the magnitude of the curls of currents at the instantaneous positions $(x_1, y_1)$ and $(x_2, y_2)$ of the two particles on a plane are exactly the same. Moreover, examining the behavior of currents in a four-dimensional space $(x_1, x_2, y_1, y_2)$, we present an evidence that the components of the currents of two particles are correlated and perform a rotational motion along closed elliptic orbits, which behavior resembles the dynamics of a Brownian gyrator [10–23]. We stress that here, however, such a dynamical behavior is observed for the like components of currents of the two particles (i.e., for the components $x_1$ and $x_2$, or $y_1$ and $y_2$), such that no net rotation of particles themselves around the origin or the centers of the traps takes place.

The paper is organized as follows: we introduce the model in section 2. Analytical expressions for the position probability density function and the probability currents in the steady-state are derived in section 3. The synchronized spinning motion of the two particles is discussed in section 4. The results of this section in the limit of a vanishingly small coupling parameter, in which limit they attain a very compact form, are presented in Appendix A. Further on, the section 5 presents an analysis of the correlated behavior of currents in a four-dimensional space. We finally conclude in section 6 with a brief recapitulation of our results.

## 2 The model

Consider a two-dimensional system with two particles - 1 and 2, which are respectively confined by two optical tweezers centered at two distinct positions. Without a lack of generality, we assume that the centers of the traps are located on the $x$-axis. We denote the positions of the centers of optical traps by vectors $-\mathbf{r} = (-x_0, 0)$ and $\mathbf{r} = (x_0, 0)$, which are both defined relative to the origin of the plane, and hence, the distance between the centers of the traps is fixed and equal to $2x_0$. In turn, the instantaneous positions of particles are specified by vectors $\mathbf{z}_1 = (x_1, y_1)$ and $\mathbf{z}_2 = (x_2, y_2)$, which are defined in the frames of reference centered at positions of the optical traps. According to such a definition, these vectors therefore determine the displacements of respective particles from the centers of two potential wells.

As shown in [34, 35], in realistic physical systems containing a solvent, the particles 1 and 2 are hydrodynamically coupled to each other - they interact through the motion of a surrounding viscous fluid. If the particles are sufficiently close to each other, the interaction

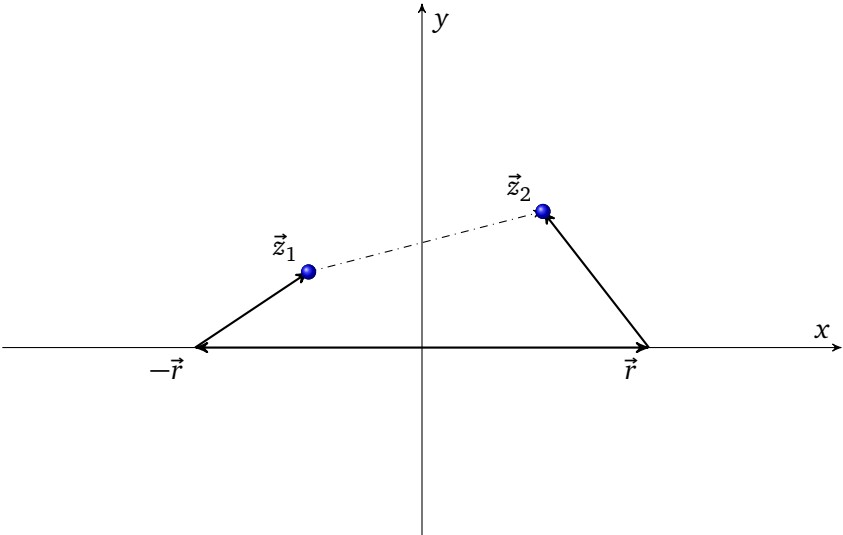

Figure 1: A geometrical set-up of the two-dimensional model under study. Filled (blue) circles denote the particles 1 and 2. The vectors $\mathbf{r}$ and $-\mathbf{r}$ determine the centers of the fixed optical traps, while the vectors $\mathbf{z}_1$ and $\mathbf{z}_2$ show the instantaneous positions of the two particles on the $(x, y)$-plane, relative to the centers of their respective traps. The distance between the centers of the traps is fixed and equal to $2x_0$.

potential $U(\rho)$ is a quadratic function of the inter-particle distance $\rho = |\mathbf{z}_1 - \mathbf{z}_2 + 2\mathbf{r}|$,

$$U(\rho) = \frac{u}{2}\rho^2, \tag{1}$$

where $u$ is the constant coupling parameter (see [34, 35]). Lastly, due to the tweezers, the particles are confined by the potential wells such that the overall potential energy $H(\mathbf{z}_1, \mathbf{z}_2)$ is given by

$$H(\mathbf{z}_1, \mathbf{z}_2) = \frac{1}{2}\gamma \mathbf{z}_1^2 + \frac{1}{2}\gamma \mathbf{z}_2^2 + \frac{1}{2}u(\mathbf{z}_1 - \mathbf{z}_2 + 2\mathbf{r})^2, \tag{2}$$

where the constant parameter $\gamma > 0$ defines the stiffness of the tweezers. We stress that in the physical situation considered in [34, 35] the form in Eq. (1) and hence, the total potential energy defined in Eq. (2), are only valid for sufficiently small values of $x_0$. For larger value of $x_0$ the hydrodynamic coupling between the particles vanishes and hence, Eq. (1) is no longer valid. Moreover, the parameter $\gamma$ should be sufficiently large such that the excursions of both particles away from the centers of their respective traps should be small, in order to ensure the validity of the form in Eq. (1). Having in mind these restrictions, we provide in what follows a formal solution of the model in Eq. (2) for arbitrary values of $\gamma > 0$ and arbitrary values of $u$, which may also attain negative values such that $u > -\gamma/2$. The meaning of the latter inequality will be made clear below.

We define next the dynamics of our model. Expanding the right-hand-side of Eq. (2) and dropping the constant term, which is irrelevant for the further analysis, we rewrite the total potential energy as

$$H(\mathbf{z}_1, \mathbf{z}_2) = \frac{1}{2}\kappa \mathbf{z}_1^2 + \frac{1}{2}\kappa \mathbf{z}_2^2 - u(\mathbf{z}_1 \cdot \mathbf{z}_2) - 2u(\mathbf{r} \cdot \mathbf{z}_1) + 2u(\mathbf{r} \cdot \mathbf{z}_2), \tag{3}$$

where the parameter $\kappa = \gamma + u$ and $(\cdot)$ denotes the scalar product. Then, we stipulate that the deviations $\mathbf{z}_1$ and $\mathbf{z}_2$ of the particles positions from the centers of their respective traps

obey a pair of coupled over-damped Langevin equations:

$$\frac{d}{dt}\mathbf{z_1}(t) = -\boldsymbol{\nabla}_1 H(\mathbf{z_1}, \mathbf{z_2}) + \boldsymbol{\xi}_1(t),$$

$$\frac{d}{dt}\mathbf{z_2}(t) = -\boldsymbol{\nabla}_2 H(\mathbf{z_1}, \mathbf{z_2}) + \boldsymbol{\xi}_2(t), \tag{4}$$

in which the symbols $\boldsymbol{\nabla}_i$ denote the gradient operators while the vectors $\boldsymbol{\xi}_i = (\xi_i^x, \xi_i^y)$, $(i = 1, 2)$ stand for statistically-independent thermal noises, with zero mean and the correlation function

$$\langle \xi_i^\alpha(t)\xi_j^\beta(t')\rangle = 2T_i\,\delta_{\alpha\beta}\,\delta_{ij}\,\delta(t-t'), \tag{5}$$

where $T_1$ and $T_2$ are the temperatures at which the particles 1 and 2 live. In the general case, $T_1 \neq T_2$, meaning that there is no unique temperature characterising the system and hence, the system does not converge to thermal equilibrium in the limit $t \to \infty$. We concentrate in what follows precisely on this out-of-equilibrium case seeking its consequences on the behavior of some observable properties.

## 3   Solution in the steady-state

Let $P_t(\mathbf{z_1}, \mathbf{z_2})$ denote the position probability density function at time $t$ and $P(\mathbf{z_1}, \mathbf{z_2})$ stand for its limiting form attained when $t \to \infty$. In this limit, the Fokker-Planck equation associated with the Langevin equations (4) has the form

$$
\begin{aligned}
0 = {}& \boldsymbol{\nabla}_1\Big[T_1\boldsymbol{\nabla}_1 P(\mathbf{z_1}, \mathbf{z_2}) + P(\mathbf{z_1}, \mathbf{z_2})\boldsymbol{\nabla}_1 H(\mathbf{z_1}, \mathbf{z_2})\Big] \\
&+ \boldsymbol{\nabla}_2\Big[T_2\boldsymbol{\nabla}_2 P(\mathbf{z_1}, \mathbf{z_2}) + P(\mathbf{z_1}, \mathbf{z_2})\boldsymbol{\nabla}_2 H(\mathbf{z_1}, \mathbf{z_2})\Big].
\end{aligned}
\tag{6}
$$

Introducing the probability currents

$$\mathbf{j}_1(\mathbf{z_1}, \mathbf{z_2}) = T_1\boldsymbol{\nabla}_1 P(\mathbf{z_1}, \mathbf{z_2}) + P(\mathbf{z_1}, \mathbf{z_2})\boldsymbol{\nabla}_1 H(\mathbf{z_1}, \mathbf{z_2}),$$

$$\mathbf{j}_2(\mathbf{z_1}, \mathbf{z_2}) = T_2\boldsymbol{\nabla}_1 P(\mathbf{z_1}, \mathbf{z_2}) + P(\mathbf{z_1}, \mathbf{z_2})\boldsymbol{\nabla}_2 H(\mathbf{z_1}, \mathbf{z_2}), \tag{7}$$

one can conveniently rewrite the above Fokker-Planck equation (6) as

$$\boldsymbol{\nabla}_1\mathbf{j}_1(\mathbf{z_1}, \mathbf{z_2}) + \boldsymbol{\nabla}_2\mathbf{j}_2(\mathbf{z_1}, \mathbf{z_2}) = 0, \tag{8}$$

which implies that the total current is conserved.

Because the total potential energy in Eq. (3) is the quadratic function of the particles' positions, the solution is evidently a Gaussian function of the form

$$P(\mathbf{z_1}, \mathbf{z_2}) = Z^{-1}\exp\Big(-\frac{1}{2}\kappa A z_1^2 - \frac{1}{2}\kappa B z_2^2 + uC(\mathbf{z_1}\cdot\mathbf{z_2}) + 2uD(\mathbf{r}\cdot\mathbf{z}_1) - 2uE(\mathbf{r}\cdot\mathbf{z}_2)\Big), \tag{9}$$

where $Z$ is a normalization constant,

$$
\begin{aligned}
Z &= \int\int d\mathbf{z}_1\, d\mathbf{z}_2\, \exp\Big(-\frac{1}{2}\kappa A z_1^2 - \frac{1}{2}\kappa B z_2^2 + uC(\mathbf{z_1}\cdot\mathbf{z_2}) + 2uD(\mathbf{r}\cdot\mathbf{z}_1) - 2uE(\mathbf{r}\cdot\mathbf{z}_2)\Big) \\
&= \frac{4\pi^2}{AB\kappa^2 - C^2 u^2}\exp\left(\frac{2x_0^2 u^2(\kappa(AE^2 + BD^2) - 2uCDE)}{AB\kappa^2 - C^2 u^2}\right),
\end{aligned}
\tag{10}
$$

and the coefficients $A$, $B$, $C$, $D$ and $E$ are to be defined. In order to determine the unknown coefficients, we first substitute Eqs. (9) and (3) into Eqs. (7), to get the following expressions for the currents

$$\mathbf{j}_1(\mathbf{z}_1, \mathbf{z}_2) = \left[ \kappa\left(1 - AT_1\right)\mathbf{z}_1 \, + \, u\left(CT_1 - 1\right)\mathbf{z}_2 \, + \, 2u\left(DT_1 - 1\right)\mathbf{r} \right] P(\mathbf{z}_1, \mathbf{z}_2), \qquad (11)$$

$$\mathbf{j}_2(\mathbf{z}_1, \mathbf{z}_2) = \left[ \kappa\left(1 - BT_2\right)\mathbf{z}_2 \, + \, u\left(CT_2 - 1\right)\mathbf{z}_1 \, - \, 2u\left(ET_2 - 1\right)\mathbf{r} \right] P(\mathbf{z}_1, \mathbf{z}_2). \qquad (12)$$

Inserting next the above expressions into the Fokker-Planck equation (8), we obtain *six* equations for *five* unknown coefficients $A$, $B$, $C$, $D$ and $E$:

$$A(1 - AT_1) - \frac{u^2}{\kappa^2} C(CT_2 - 1) = 0, \qquad (13)$$

$$B(1 - BT_2) - \frac{u^2}{\kappa^2} C(CT_1 - 1) = 0, \qquad (14)$$

$$A(1 - 2CT_1) + B(1 - 2CT_2) = -2C, \qquad (15)$$

$$\frac{u}{\kappa}(E + C(1 - 2ET_2)) + (A + D - 2ADT_1) = 0, \qquad (16)$$

$$\frac{u}{\kappa}(D + C(1 - 2DT_1)) + (B + E - 2BET_2) = 0, \qquad (17)$$

$$\kappa(1 - AT_1) + \kappa(1 - BT_2) + 2\frac{u^2 x_0^2}{\kappa}(D(DT_1 - 1) + E(ET_2 - 1)) = 0. \qquad (18)$$

From Eqs. (13) to (15), we readily find that $A$, $B$, and $C$ obey

$$A = \frac{1}{T_1} + \frac{u^2(T_1^2 - T_2^2)}{(4\kappa^2 T_1 T_2 + u^2(T_1 - T_2)^2)T_1}, \qquad (19)$$

$$B = \frac{1}{T_2} - \frac{u^2(T_1^2 - T_2^2)}{(4\kappa^2 T_1 T_2 + u^2(T_1 - T_2)^2)T_2}, \qquad (20)$$

$$C = \frac{2\kappa^2(T_1 + T_2)}{4\kappa^2 T_1 T_2 + u^2(T_1 - T_2)^2}. \qquad (21)$$

Then, Eqs. (16) and (17) give

$$D = \frac{4\kappa^2 T_2 + 2\kappa u(T_1 - T_2)}{4\kappa^2 T_1 T_2 + u^2(T_1 - T_2)^2}, \qquad (22)$$

$$E = \frac{4\kappa^2 T_1 + 2\kappa u(T_2 - T_1)}{4\kappa^2 T_1 T_2 + u^2(T_1 - T_2)^2}. \qquad (23)$$

Note that for the above solution the Eq. (18) holds as an identity, so that the system of equations (13) to (18) is not overdetermined, and also that the coefficients are actually independent of the distance $2x_0$ between the centers of the optical traps, which enters only in Eq. (18). We are now equipped with all necessary ingredients to find explicit expressions for the normalisation $Z$ and the probability currents. Inserting Eqs. (19) to (23) into Eq. (10), we have

$$Z = \frac{\pi^2((T_2 - T_1)^2 u^2 + 4T_1 T_2 \kappa^2)}{\kappa^4 - \kappa^2 u^2} \exp\left( \frac{8x_0^2(T_1 + T_2)u^2 \kappa^2}{(u + \kappa)((T_2 - T_1)^2 u^2 + 4T_1 T_2 \kappa^2)} \right). \qquad (24)$$

Note that the normalization constant $Z$, Eq. (24), is bounded when $\gamma > 0$ and positive whenever $\kappa^2 > u^2$, in which case the system is stable. The latter inequality is realised when

$u > -\gamma/2$, (recall that $\kappa = u + \gamma$), which explains the above imposed constraint (see the paragraph below Eq. (2)). Note, as well, that the parameter $u$ can therefore be negative meaning that our analysis is also valid for the systems in which the particles (sufficiently weakly) repel each other. In turn, the probability currents are given explicitly by

$$
\begin{aligned}
\mathbf{j_1}(\mathbf{z_1}, \mathbf{z_2}) &= \frac{u(T_2 - T_1)P(\mathbf{z_1}, \mathbf{z_2})}{4\kappa^2 T_1 T_2 + u^2(T_1 - T_2)^2} \\
&\times \Big[ \kappa u(T_1 + T_2)\mathbf{z_1} - ((T_2 - T_1)u^2 + 2\kappa^2 T_1)\mathbf{z_2} + 2((T_2 - T_1)u^2 + 2u\kappa T_1)\mathbf{r} \Big], \\
\mathbf{j_2}(\mathbf{z_1}, \mathbf{z_2}) &= \frac{u(T_1 - T_2)P(\mathbf{z_1}, \mathbf{z_2})}{4\kappa^2 T_1 T_2 + u^2(T_1 - T_2)^2} \\
&\times \Big[ \kappa u(T_1 + T_2)\mathbf{z_2} - ((T_1 - T_2)u^2 + 2\kappa^2 T_2)\mathbf{z_1} + 2((T_1 - T_2)u^2 + 2u\kappa T_2)\mathbf{r} \Big].
\end{aligned}
\tag{25}
$$

Therefore, in out-of-equilibrium conditions (i.e. for $T_1 \neq T_2$), and also for a non-zero coupling between the two particles (i.e. when $u \neq 0$), there exist non-vanishing probability currents in the steady-state. Below we discuss some remarkable features of the dynamical behavior, which originate from this latter circumstance.

## 4  Synchronous spinning of particles

Our aim now is to demonstrate that the probability currents possess a non-zero curl, i.e., the velocity field undergoes a circulation. The curls $S_1(\mathbf{z_1}, \mathbf{z_2})$ and $S_2(\mathbf{z_1}, \mathbf{z_2})$ are formally defined as the circulation density at "point" $(\mathbf{z_1}, \mathbf{z_2})$ of the field, i.e., $S_1(\mathbf{z_1}, \mathbf{z_2}) = (\boldsymbol{\nabla}_1 \times \mathbf{j_1}) \cdot \hat{\mathbf{k}}$ and $S_2(\mathbf{z_1}, \mathbf{z_2}) = (\boldsymbol{\nabla}_2 \times \mathbf{j_2}) \cdot \hat{\mathbf{k}}$, where $\hat{\mathbf{k}}$ is the unit vector in the direction orthogonal to the $(x, y)$-plane and the symbol $(\times)$ denotes the vector product. Taking advantage of the above equations (25), we readily find that the curls are given explicitly by

$$
S_1(\mathbf{z_1}, \mathbf{z_2}) = \frac{2u\lambda\kappa(T_1 - T_2)[2x_0 u(y_1 + y_2) + (x_2 y_1 - x_1 y_2)(u + \kappa)]}{(T_1 - T_2)^2 u^2 + 4T_1 T_2 \kappa^2} P(\mathbf{z_1}, \mathbf{z_2}),
\tag{26}
$$

and

$$
S_2(\mathbf{z_1}, \mathbf{z_2}) = \frac{2u\lambda\kappa(T_1 - T_2)[2x_0 u(y_1 + y_2) + (x_2 y_1 - x_1 y_2)(u + \kappa)]}{(T_1 - T_2)^2 u^2 + 4T_1 T_2 \kappa^2} P(\mathbf{z_1}, \mathbf{z_2}).
\tag{27}
$$

Remarkably, the curls $S_1(\mathbf{z_1}, \mathbf{z_2})$ and $S_2(\mathbf{z_1}, \mathbf{z_2})$ are a) both non-zero in out-of-equilibrium conditions and for $u \neq 0$ and moreover, b) are exactly equal to each other at any point $(\mathbf{z_1}, \mathbf{z_2})$. First, this implies that if the particles were to have a finite-size, the field will create a net torque on each particle such that it will steadily spin about its center of mass. Second, such a spinning motion of the two particles will be completely synchronized in the sense that both the sign and the magnitude of the curls $S_1(\mathbf{z_1}, \mathbf{z_2})$ and $S_2(\mathbf{z_1}, \mathbf{z_2})$ are exactly the same. For $u > 0$ and $T_1 > T_2$, the curls will be positive if the coordinates of particles' displacements from the centers of the optical traps obey

$$
2x_0 u(y_1 + y_2) + (x_2 y_1 - x_1 y_2)(u + \kappa) > 0,
\tag{28}
$$

and will be less than zero, otherwise. When Eq. (28) becomes an equality, the curls vanish such that the spinning motion stops. This happens, in particular, when both particles appear at the centers of their respective optical traps.

The curl of either of the currents, e.g., of $\mathbf{j}_1(\mathbf{z_1}, \mathbf{z_2})$, integrated over all possible positions of either of the particles vanishes, i. e.,

$$\int d\mathbf{z}_1 \, S_1(\mathbf{z_1}, \mathbf{z_2}) = \int d\mathbf{z}_2 \, S_1(\mathbf{z_1}, \mathbf{z_2}) = 0 \,. \tag{29}$$

By symmetry, the same is true for $S_2(\mathbf{z_1}, \mathbf{z_2})$. It seems interesting, however, to determine a property which does not vanish when it is integrated over positions of the particles. To this end, we consider the absolute values of the curls integrated over all possible positions of one of the particles with the second one being fixed at the center of the optical trap:

$$\begin{aligned}
\langle |S_1| \rangle &= \int d\mathbf{z_2} |S_1(0, \mathbf{z_2})| \,, \\
\langle |S_2| \rangle &= \int d\mathbf{z_1} |S_1(\mathbf{z_1}, 0)| \,.
\end{aligned} \tag{30}$$

Inserting our expressions (26) and (27) into Eqs. (30) and performing the integrations, we find after some algebra

$$\begin{aligned}
\langle |S_1| \rangle &= \frac{4\kappa(\gamma + 2u)x_0 u^2 \gamma^2 |(T_2 - T_1)|}{\pi^{3/2} \sigma_1^3} \sqrt{\frac{(u+\gamma)}{(T_1 + T_2)^2 u^2 + 8T_1 T_2 u\gamma + 4T_1 T_2 \gamma^2}} \\
&\quad \times \exp\left(\frac{-4(u+\gamma)\gamma x_0^2 u^2}{(\gamma + 2u)\sigma_1^2}\right), \\
\langle |S_2| \rangle &= \frac{4\kappa(\gamma + 2u)x_0 u^2 \gamma^2 |(T_2 - T_1)|}{\pi^{3/2} \sigma_2^3} \sqrt{\frac{(u+\gamma)}{(T_1 + T_2)^2 u^2 + 8T_1 T_2 u\gamma + 4T_1 T_2 \gamma^2}} \\
&\quad \times \exp\left(\frac{-4(u+\gamma)\gamma x_0^2 u^2}{(\gamma + 2u)\sigma_2^2}\right),
\end{aligned} \tag{31}$$

where we have used the shortened notations

$$\sigma_1^2 \equiv (T_1 + T_2)u^2 + 4T_1 u\gamma + 2T_1 \gamma^2 \,, \tag{32}$$

$$\sigma_2^2 \equiv (T_1 + T_2)u^2 + 4T_2 u\gamma + 2T_2 \gamma^2 \,. \tag{33}$$

Hence, the integrated absolute values of the curls $\langle |S_1| \rangle$ and $\langle |S_2| \rangle$ do not vanish when the product $x_0 u(T_2 - T_1) \neq 0$. This occurs when the following three conditions are simultaneously met: the temperatures are different, the coupling between particles and also the distance between the two optical centers are not equal to zero. The non zero values of $\langle |S_1| \rangle$ and $\langle |S_2| \rangle$ imply that there exists a synchronized motion of particles in the stationary state.

It may be also instructive to consider the ratio of $\langle |S_1| \rangle$ and $\langle |S_2| \rangle$. From Eqs. (31) we find

$$\frac{\langle S_1 \rangle}{\langle S_2 \rangle} = \exp\left(\frac{8\kappa x_0^2 \lambda^2 u^2 (T_1 - T_2)}{4\kappa^4 T_1 T_2 - u^4 (T_1 - T_2)^2 + \kappa^2 u^2 (T_1 - T_2)^2}\right) \left|\frac{u^2(T_2 - T_1) + 2\kappa^2 T_2}{u^2(T_1 - T_2) - 2\kappa^2 T_1}\right|^{3/2}. \tag{34}$$

Expanding the latter expression in powers of the coupling parameter $u$, we have

$$\frac{\langle |S_1| \rangle}{\langle |S_2| \rangle} = \left(\frac{T_2}{T_1}\right)^{3/2} + \frac{u^2}{4\kappa^2} \frac{(T_1 - T_2)}{T_1 T_2} \left(\frac{T_2}{T_1}\right)^{3/2} \left(8\kappa x_0^2 + 3T_1 + 3T_2\right) + O\left(u^3\right), \tag{35}$$

where the symbol $O\left(u^3\right)$ signifies that the omitted correction terms are proportional to $u^3$. Equation (35) implies that $\langle |S_1| \rangle$ and $\langle |S_2| \rangle$ can be disproportionally different, if the temperatures are very different. In particular, $\langle |S_1| \rangle$ can be much larger than $\langle |S_2| \rangle$ if $T_2 \ll T_1$.

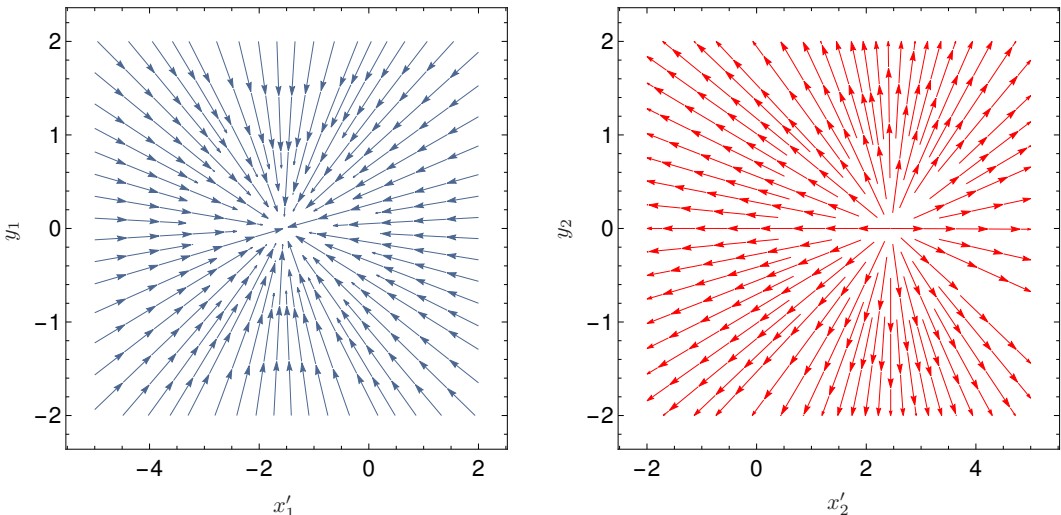

Figure 2: Streamplots of the probability currents $\mathbf{j_1}\big((x_1', y_1), \mathbf{z_2} = 0\big)$ (left panel) - a vector with components $j_{1,x}\big((x_1', y_1), (0,0)\big)$ and $j_{1,y}\big((x_1', y_1), (0,0)\big)$, and $\mathbf{j_2}\big(\mathbf{z_1} = 0, (\mathbf{x_2}, \mathbf{y_2})\big)$ (right panel) - a vector with components $j_{2,x}\big((0,0), (x_2', y_2)\big)$ and $j_{2,y}\big((0,0), (x_2', y_2)\big)$. Here, $x_1' = x_1 - x_0$ and $x_2' = x_2 + x_0$ are the $x$-coordinates in the laboratory reference frame (see Fig. 1). The values of the parameters are: $\gamma = 1$, $u = 1/2$, $x_0 = 1$, $T_1 = 1$ and $T_2 = 2$.

## 5 Correlated behavior of currents

In this section we discuss an emerging cooperative behavior of the probability currents defined in Eqs. (25). To this end, we study the correlations between the components of the probability currents in a four-dimensional space $(x_1, x_2, y_1, y_2)$, accessing them via the streamplots of the projections on different planes.

Figure 2 shows streamplots of the currents $\mathbf{j_1}\big((x_1, y_1), \mathbf{z_2} = 0\big)$, i.e., the current associated with the particle 1 with the particle 2 being fixed at the center of its optical trap, and $\mathbf{j_2}\big(\mathbf{z_1} = 0, (x_2, y_2)\big)$ - the current associated with particle 2 with the particle 1 being fixed at the center of its trap. In this and the subsequent figure we choose the following values of the parameters: $\gamma = 1$, $u = 1/2$, $x_0 = 1$, $T_1 = 1$ and $T_2 = 2$. We observe that the streamplot of $\mathbf{j_1}\big((x_1, y_1), \mathbf{z_2} = 0\big)$ consists of curves which travel from infinity to some fixed point, while the one for $\mathbf{j_2}\big(\mathbf{z_1} = 0, (x_2, y_2)\big)$ consists of curves which starts from some point and travel to infinity. In both cases the curves are not closed, as it happens for the Brownian gyrator (see e.g. [19]); the reason for such a behavior is that both components of each current are living at the same temperature.

We consider next the behavior of the $x$-components of the two currents, which are subject to two *different* temperatures, as well as the behavior of the $y$-components. In the left panel in Fig. 3 we present a streamplot of the vector with components $\big(j_{1,x}(\mathbf{z_1}, \mathbf{z_2}), j_{2,x}(\mathbf{z_1}, \mathbf{z_2})\big)$. Observe that the behavior is completely different from the one presented in Fig. 2 - the $x$-components of the two currents perform a circulation on the $(x_1, x_2)$-plane along closed elliptic curves. Essentially the same behavior, which reveals an emerging cooperativity, is exhibited by the $y$-components of the two currents as depicted on the left panel in Fig. 3. This is precisely what was previously observed for the Brownian gyrator model on a plane with different temperatures along the two Cartesian directions. Here, however, neither of the particles themselves performs a gyration along some point on a plane but rather the components of the

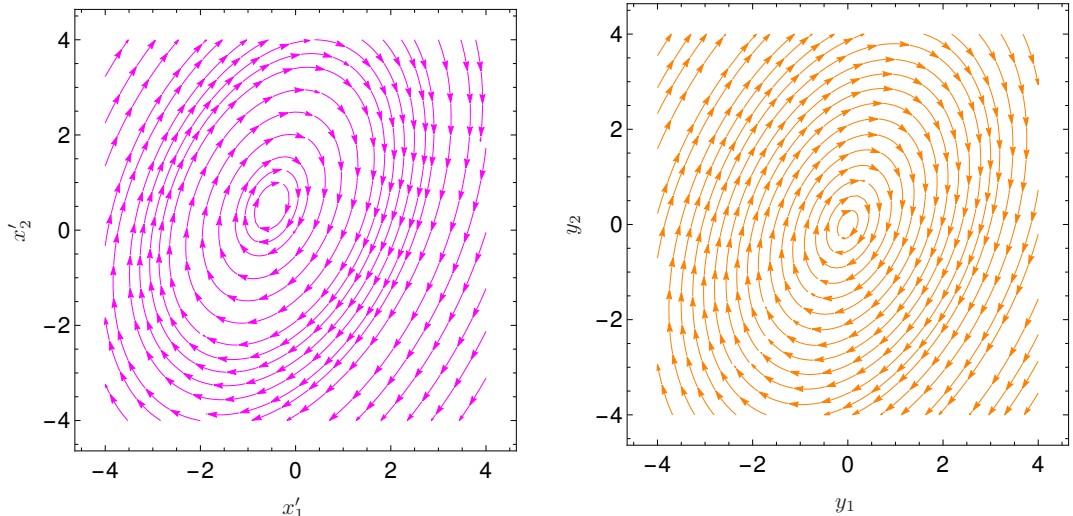

Figure 3: Streamplots of the vectors $\big(j_{1,x}(\mathbf{z_1},\mathbf{z_2}),j_{2,x}(\mathbf{z_1},\mathbf{z_2})\big)$ with fixed $y_1 = 1$, $y_2 = 2$ (left panel), and $\big(j_{1,y}(\mathbf{z_1},\mathbf{z_2}),j_{2,y}(\mathbf{z_1},\mathbf{z_2})\big)$ with $x_1' = 1$, $x_2' = 2$ (right panel), where $x_1' = x_1 - x_0$ and $x_2' = x_2 + x_0$ are the $x$-coordinates in the absolute reference frame (see Fig. 1). Here, $\gamma = 1$, $u = 1/2$, $x_0 = 1$, $T_1 = 1$ and $T_2 = 2$.

currents of two particles circulate along closed orbits in a correlated manner.

To further characterise the circulation of the probability currents on the planes $(x_1, x_2)$ and $(y_1, y_2)$, evidenced in Fig. 3, we evaluate below several additional properties. These are a) the values of the curls of the probability currents at positions of the optical traps, b) the mean angular momenta and c) the mean angular velocities of the circulation.

a) Consider first the curls of the probability currents on the planes $(x_1, x_2)$ and $(y_1, y_2)$ defined as

$$S_x\big(\mathbf{z_1},\mathbf{z_2}\big) = \frac{\partial}{\partial x_1} j_{2,x}\big(\mathbf{z_1},\mathbf{z_2}\big) - \frac{\partial}{\partial x_2} j_{1,x}\big(\mathbf{z_1},\mathbf{z_2}\big), \tag{36}$$

and

$$S_y\big(\mathbf{z_1},\mathbf{z_2}\big) = \frac{\partial}{\partial y_1} j_{2,y}\big(\mathbf{z_1},\mathbf{z_2}\big) - \frac{\partial}{\partial y_2} j_{1,y}\big(\mathbf{z_1},\mathbf{z_2}\big). \tag{37}$$

Taking advantage of Eqs. (25), we find that the values of these curls at the locations of the centers of the optical traps are given explicitly by

$$\begin{aligned}
S_x(\{0, y_1\}, \{0, y_2\}) &= \frac{2u\kappa\Delta(T_1 - T_2)}{((T_1 - T_2)^2 u^2 + 4T_1 T_2 \kappa^2)^2} P\big(\{0, y_1\}, \{0, y_2\}\big), \\
S_y(\{x_1, 0\}, \{x_2, 0\}) &= \frac{2u\kappa^2(T_2^2 - T_1^2)}{(T_1 - T_2)^2 u^2 + 4T_1 T_2 \kappa^2} P\big(\{x_1, 0\}, \{x_2, 0, \}\big),
\end{aligned} \tag{38}$$

where

$$\begin{aligned}
\Delta = {}& 8x_0^2(T_1 - T_2)^2 u^4 - (T_1 - T_2)^2 u^2(T_1 + T_2 + 16x_0^2 u)\kappa \\
& + 16x_0^2(T_1^2 + T_2^2)u^2\kappa^2 - 4T_1 T_2(T_1 + T_2)\kappa^3.
\end{aligned} \tag{39}$$

Note that there is no symmetry between the expressions in the first and the second line in Eqs. (38), which is due to the fact that the centers of both optical traps are located on the $x$-axis.

b) The angular momentum (per unit mass) for the rotation of the probability current on the $(x_1, x_2)$-plane is defined as

$$L_{x_1,x_2} = x_1 j_{2,x} - x_2 j_{1,x} \,, \tag{40}$$

such that its averaged value is given by

$$\left\langle L_{x_1,x_2} \right\rangle = \int \int d\mathbf{z_1} d\mathbf{z_2} \left( x_1 j_{2,x} - x_2 j_{1,x} \right) . \tag{41}$$

Similarly, the angular momentum for the rotation on the $(y_1, y_2)$-plane and its averaged value follow

$$L_{y_1,y_2} = y_1 j_{2,y} - y_2 j_{1,y} \,, \tag{42}$$

and

$$\left\langle L_{y_1,y_2} \right\rangle = \int \int d\mathbf{z_1} d\mathbf{z_2} \left( y_1 j_{2,y} - y_2 j_{1,y} \right) . \tag{43}$$

Using our Eqs. (25) and performing the integrals, we find that the averaged values of the angular momenta on the $(x_1, x_2)$ and $(y_1, y_2)$ are exactly equal to each other and are both given by a very simple expression

$$\left\langle L_{x_1,x_2} \right\rangle = \left\langle L_{y_1,y_2} \right\rangle = \frac{u(T_2 - T_1)}{\kappa} \,. \tag{44}$$

The equality of both averaged angular momenta is rather surprising in view of the fact that the values of the curls on these planes are very different - see Eqs. (38). The averaged angular momenta defined in Eq. (44) are depicted on the left panel in Fig. 4 as functions of the coupling parameter $u$, (recall that $\kappa = u + \gamma$). The prediction in Eq. (44) is confirmed by numerical simulations by using an Euler-Maruyama method [38] with time step $\delta t = 2 \times 10^{-3}$ and the total elapsed time $t_f = 2000$. Each simulation result (given by filled circles in Fig. 4) is an average performed over 48 to 96 independent runs.

c) Lastly, we calculate the averaged angular velocities $\left\langle \omega_{x_1,x_2} \right\rangle$ and $\left\langle \omega_{y_1,y_2} \right\rangle$ of circulations of the probability currents on the planes $(x_1, x_2)$ and $(y_1, y_2)$ which are defined as

$$
\begin{aligned}
\left\langle \omega_{x_1,x_2} \right\rangle &= \int \int d\mathbf{z_1} d\mathbf{z_2} \left( \frac{x_1 j_{2,x} - x_2 j_{1,x}}{x_1^2 + x_2^2} \right) , \\
\left\langle \omega_{y_1,y_2} \right\rangle &= \int \int d\mathbf{z_1} d\mathbf{z_2} \left( \frac{y_1 j_2^y - y_2 j_1^y}{y_1^2 + y_2^2} \right) .
\end{aligned}
\tag{45}
$$

Performing the integrals, we eventually find

$$
\begin{aligned}
\left\langle \omega_{x_1,x_2} \right\rangle &= u(T_2 - T_1) \sqrt{\frac{\kappa^2 - u^2}{(T_1 - T_2)^2 u^2 + 4 T_1 T_2 \kappa^2}} \\
&\quad \times \exp\left( -\frac{8 x_0^2 (T_1 + T_2) u^2 \kappa^2}{(u + \kappa)((T_2 - T_1)^2 u^2 + 4 T_1 T_2 \kappa^2)} \right) , \\
\left\langle \omega_{y_1,y_2} \right\rangle &= u(T_2 - T_1) \sqrt{\frac{\kappa^2 - u^2}{(T_1 - T_2)^2 u^2 + 4 T_1 T_2 \kappa^2}} \,.
\end{aligned}
\tag{46}
$$

Equation (46) implies that the ratio of the averaged angular velocities obeys

$$\frac{\left\langle \omega_{x_1,x_2} \right\rangle}{\left\langle \omega_{y_1,y_2} \right\rangle} = \exp\left( -\frac{8 x_0^2 (T_1 + T_2) u^2 \kappa^2}{(u + \kappa)((T_2 - T_1)^2 u^2 + 4 T_1 T_2 \kappa^2)} \right) < 1 \,, \tag{47}$$

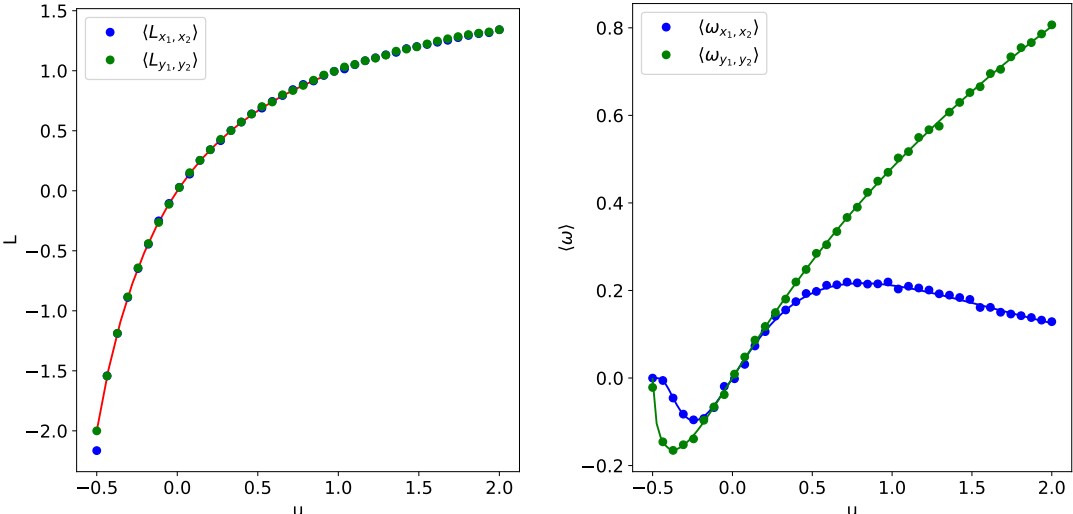

Figure 4: Left panel: Averaged angular momenta $\langle L_{x_1,x_2} \rangle$ and $\langle L_{y_1,y_2} \rangle$ as functions of the coupling parameter $u$. Numerical results are given by the full circles, while the analytical expression (44) is given by the solid curve. Right panel: Averaged angular velocities $\langle \omega_{x_1,x_2} \rangle$ and $\langle \omega_{y_1,y_2} \rangle$ as functions of $u$. Numerical results are denoted by full circles and the analytical expressions (46) are denoted by solid curves. In both panels $\gamma = 1$, $T_1 = 1$ and $T_2 = 3$.

i.e., this ratio is always less than unity, despite the fact that the averaged angular momenta and correspondingly, the torques are equal to each other (see Eq. (44)). Consequently, the averaged angular velocity for the rotations on the $(y_1, y_2)$-plane is always greater than the one for the rotations on the $(x_1, x_2)$-plane, for an arbitrary sign of the coupling parameter $u$.

Overall, $\langle \omega_{y_1,y_2} \rangle$ and $\langle \omega_{x_1,x_2} \rangle$ are non-monotonic functions of the parameter $u$ with a minimum attained at some $u = u^* < 0$. For large positive values of $u$ the behavior of $\langle \omega_{y_1,y_2} \rangle$ and $\langle \omega_{x_1,x_2} \rangle$ is markedly different: $\langle \omega_{y_1,y_2} \rangle$ diverges in proportion to a square-root of $u$:

$$\langle \omega_{y_1,y_2} \rangle \simeq \frac{T_2 - T_1}{T_2 + T_1} \sqrt{\gamma u}, \tag{48}$$

while $\langle \omega_{x_1,x_2} \rangle$ attains a maximal value when you $u$ approaches

$$u = \frac{(T_1 - T_2)^2}{8x_0^2(T_1 + T_2)}, \tag{49}$$

and then *decreases* exponentially,

$$\langle \omega_{x_1,x_2} \rangle \simeq \frac{T_2 - T_1}{T_2 + T_1} \sqrt{\gamma u} \exp\left( -\frac{4x_0^2(T_1 + T_2)u}{(T_2 - T_1)^2} \right). \tag{50}$$

The behavior of $\langle \omega_{y_1,y_2} \rangle$ and $\langle \omega_{x_1,x_2} \rangle$ as functions of $u$ is depicted on the right panel in Fig. 4 together with the results of numerical simulations which confirm our analytical predictions.

# 6 Conclusion

To conclude, we presented here a detailed theoretical analysis of an out-of-equilibrium dynamics of two interacting, randomly moving particles in a two-dimensional system, which

was realised experimentally in [34, 35]. More specifically, the experimental set-up in these references consisted of a disc-shaped shallow cell filled with a solvent and containing two suspended micrometer-sized beads, each being held by its own optical tweezer. One of the tweezers was subject to an additional, externally-imposed noise such that the particle held by this very tweezer lived at an effectively different temperature as compared to the other one. Due to the presence of a solvent, the particles were coupled by hydrodynamic interactions.

References [34, 35] focused on the behavior of the effective heat fluxes between the two beads in the out-of-equilibrium state with unequal temperatures and developed both experimental and theoretical analyses. On the theoretical side, the dynamics was framed in terms of two coupled over-damped Langevin equations with effective harmonic interactions between the particles, the parameters of which were deduced from the Rotne-Prager diffusion tensor. It was demonstrated that the heat fluxes obey, e.g., an exchange fluctuation theorem which result was confirmed both experimentally and theoretically, with a very good agreement between the two approaches. In turn, it proved directly the validity of the theoretical description based on the Langevin dynamics.

On the other hand, the theoretical analysis in [34, 35] was based on the assumption that the stochastic dynamics of the two particles can be viewed as an effectively *one-dimensional* process that evolves along the line connecting the centers of two tweezers. Here, we addressed a conceptually important question what physical effects can be potentially overlooked due to such an assumption. To this end, we formulated and analysed essentially the same model but with two particles evolving on a *plane,* which is in fact closer to the actual experimental set-up.

We have shown that, indeed, a reduction of the dynamics to a one-dimension misses some rather spectacular effects. We demonstrated that in case when the temperatures at which the particles live are different, the system reaches a steady-state with non-zero probability currents which possess non-zero curls. As a consequence, in such a system the particles are continuously spinning around their centers of mass in a completely synchronized way - the curls of currents at the instantaneous positions of two particles have the same magnitude and sign. Further on, our analysis revealed emerging correlations between the probability currents. In particular, we realised that the $x$- components (and also the $y$-components) of the currents undergo a rotational motion along closed elliptic orbits.

## Acknowledgments

The authors wish to thank Luca Peliti for many helpful discussions.

## A  Small coupling limit

In this appendix we focus on the behavior in the limit of a vanishingly small coupling parameter $u$, in which case our results attain very simple forms. For $u \to 0$, Eqs. (11) and (12) become

$$\mathbf{j}_1(\mathbf{z_1}, \mathbf{z_2}) \simeq u \frac{T_1 - T_2}{2T_2} \mathbf{z_2}\, P(\mathbf{z_1}, \mathbf{z_2}), \tag{A.1}$$

$$\mathbf{j}_2(\mathbf{z_1}, \mathbf{z_2}) \simeq u \frac{T_2 - T_1}{2T_1} \mathbf{z_1}\, P(\mathbf{z_1}, \mathbf{z_2}), \tag{A.2}$$

and the probability density function $P(\mathbf{z_1}, \mathbf{z_2})$ in Eq. (9) attains the form

$$P(\mathbf{z_1}, \mathbf{z_2}) \simeq Z^{-1} \exp\left(-\frac{\gamma}{2T_1} z_1^2 - \frac{\gamma}{2T_2} z_2^2 + u \frac{(T_1 + T_2)}{2T_1 T_2}(\mathbf{z_1} \cdot \mathbf{z_2}) + \frac{2u}{T_1}(\mathbf{r} \cdot \mathbf{z_1}) - \frac{2u}{T_2}(\mathbf{r} \cdot \mathbf{z_2})\right), \tag{A.3}$$

with

$$Z \simeq \frac{4\pi^2 T_1 T_2}{\gamma^2}. \tag{A.4}$$

Using Eqs. (A.1) and (A.2), one readily calculates the curls of the probability currents to get

$$S_1(\mathbf{z_1}, \mathbf{z_2}) = \boldsymbol{\nabla}_1 \times \mathbf{j}_1 = \frac{\partial}{\partial x_1} j_{1,y} - \frac{\partial}{\partial y_1} j_{1,x}$$

$$= u \frac{T_1 - T_2}{2T_2} \left( y_2 \frac{\partial}{\partial x_1} P(\mathbf{z_1}, \mathbf{z_2}) - x_2 \frac{\partial}{\partial y_1} P(\mathbf{z_1}, \mathbf{z_2}) \right)$$

$$\simeq u \frac{T_1 - T_2}{2T_1 T_2} \left( 2u (\mathbf{r} \times \mathbf{z}_2) - \gamma (\mathbf{z}_1 \times \mathbf{z}_2) \right) P(\mathbf{z_1}, \mathbf{z_2}). \tag{A.5}$$

The above expression simplifies considerably in case when the particle 1 resides in the center of its optical trap, i.e. $\mathbf{z}_1 = 0$,

$$S_1(0, \mathbf{z_2}) \simeq u^2 \frac{(T_1 - T_2)}{T_1 T_2} (\mathbf{r} \times \mathbf{z}_2) P(0, \mathbf{z_2}), \tag{A.6}$$

or explicitly,

$$S_1(0, \mathbf{z_2}) \simeq \frac{u^2 \gamma^2}{4\pi^2} \frac{(T_1 - T_2)}{T_1^2 T_2^2} (\mathbf{r} \times \mathbf{z}_2) \exp\left( -\frac{\gamma}{2T_2} z_2^2 - \frac{2u}{T_2} (\mathbf{r} \cdot \mathbf{z}_2) \right). \tag{A.7}$$

Equation (A.7) shows in a transparent way that the curl of the probability vanishes when the temperatures $T_1$ and $T_2$ are equal to each other, and also when the coupling parameter $u$ or the stiffness $\gamma$ of the optical trap are equal to zero.

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
