# Peer review of "Out-of-equilibrium dynamics of two interacting optically-trapped particles"

_SciPost Physics Core, doi:SciPost Phys. Core 6, 056 (2023)_

## Round 1 · Referee Report · Anonymous (Referee 1) · 2023-5-9

Strengths

- Clearly defined problem and solid solution
- Of experimental relevance
- Well-written and pedagogical
- Appears to be sound and correct

Weaknesses

- Extends earlier work, rather than being completely original and novel.

Report

The authors investigate theoretically the non-equilibrium dynamics of two interacting particles that move randomly in a plane under the influence of different temperatures. Such a setup has recently been realized experimentally using optical tweezers with effective temperatures implemented by additional noise. The authors find that the particles spin around their centers-of-mass in a synchronised way. Compared to earlier works that focused on a one-dimensional setup, the authors now consider a plane and unravel the dynamical behavior that can occur in two dimensions.

I like the paper and believe that it can be published as it is. The problem is clearly laid out, and the solution seems correct and sound. The findings may also be of relevance to current experimental actitivies.

I just have a few minor comments for the authors to consider:

- I wonder if one can interpret the normalization factor in Eqs. (10,24), with the suggestive letter Z, as a non-equilibrium partition function? If so, would it be possible to extract further information from this quantity, for example, by going beyond average values and consider fluctuations as well?

- The authors might want to proof-read the manuscript one last time; e.g. in the caption of Fig. 4, there is a typo in "averaged".

---

## Editorial Decision

published